# Comparative transcriptomics in three *Methylophilaceae* species uncover different strategies for environmental adaptation

Alexey Vorobev[1], David A.C. Beck[2], Marina G. Kalyuzhnaya[3], Mary E. Lidstrom[4] and Ludmila Chistoserdova[1]

[1] Department of Chemical Engineering, University of Washington, Seattle, WA, USA
[2] Department of Chemical Engineering and eScience Institute, University of Washington, Seattle, WA, USA
[3] Department of Microbiology, University of Washington, Seattle, WA, USA
[4] Departments of Chemical Engineering and Microbiology, University of Washington, Seattle, WA, USA

Corresponding author
Ludmila Chistoserdova,
milachis@uw.edu

## ABSTRACT

We carried out whole transcriptome analysis of three species of *Methylophilaceae*, *Methylotenera mobilis*, *Methylotenera versatilis* and *Methylovorus glucosotrophus*, in order to determine which metabolic pathways are actively transcribed in cultures grown in laboratory on C1 substrates and how metabolism changes under semi-*in situ* conditions. Comparative analyses of the transcriptomes were used to probe the metabolic strategies utilized by each of the organisms in the environment. Our analysis of transcript abundance data focused on changes in expression of methylotrophy metabolic modules, as well as on identifying any functional modules with pronounced response to *in situ* conditions compared to a limited set of laboratory conditions, highlighting their potential role in environmental adaptation. We demonstrate that transcriptional responses to environmental conditions involved both methylotrophy and non-methylotrophy metabolic modules as well as modules responsible for functions not directly connected to central metabolism. Our results further highlight the importance of XoxF enzymes that were previously demonstrated to be highly expressed *in situ* and proposed to be involved in metabolism of methanol by *Methylophilaceae*. At the same time, it appears that different species employ different homologous Xox systems as major metabolic modules. This study also reinforces prior observations of the apparent importance of the methylcitric acid cycle in the *Methylotenera* species and its role in environmental adaptation. High transcription from the respective gene clusters and pronounced response to *in situ* conditions, along with the reverse expression pattern for the ribulose monophosphate pathway that is the major pathway for carbon assimilation in laboratory conditions suggest that a switch in central metabolism of *Methylotenera* takes place in response to *in situ* conditions. The nature of the metabolite(s) processed via this pathway still remains unknown. Of the functions not related to central metabolism, flagellum and fimbria synthesis functions appeared to be of significance for environmental adaptation, based on their high abundance and differential expression. Our data demonstrate that, besides shared strategies, the organisms employed in

this study also utilize strategies unique to each species, suggesting that the genomic divergence plays a role in environmental adaptation.

## INTRODUCTION

Bacteria of the family *Methylophilaceae* are ubiquitous in natural environments, with the exception of extreme environments, and are found in fresh and saline waters, soils, air, industrial waste-water treatment reactors etc., pointing to the environmental importance of this group (*Chistoserdova, 2011a*). Four official genera within *Methylophilaceae* have been described, *Methylophilus, Methylobacillus, Methylovorus* (*Doronina, Ivanova & Trotsenko, 2005*) and more recently *Methylotenera* (*Kalyuzhnaya et al., 2006*; *Kalyuzhnaya et al., 2012*). However, marine *Methylophilaceae* are sufficiently divergent to warrant naming at least one more genus, i.e., they are more than 5% divergent at the 16S rRNA gene sequence level from any *Methylophilaceae* classified to genus level (*Giovannoni et al., 2008*; *Huggett, Hayakawa & Rappé, 2012*). The marine *Methylophilaceae* also possess extremely small genomes compared to terrestrial *Methylophilaceae*, apparently as a result of massive gene loss (*Giovannoni et al., 2008*).

Some *Methylophilaceae* are very easy to cultivate, and these (mostly *Methylophilus* and *Methylobacillus* species) have served for decades as models for studying the biology of methylotrophs utilizing the ribulose monophosphate pathway (RuMP) for formaldehyde assimilation (*Anthony, 1982*). Based on these studies, *Methylophilaceae* have been assumed to be fast growers tolerating high concentrations of C1 substrates (i.e., substrates not containing carbon-carbon bonds), typically methanol and methylamine, resulting in high biomass yields (*Baev et al., 1992*). In fact, these properties have been exploited in commercial production of an animal feed protein from the biomass of *Methylophilus methylotrophus* (*Anthony, 1982*). The growth characteristics of these methylotrophs have been correlated with the presence and high activities of, respectively, methanol dehydrogenase (MDH) and methylamine dehydrogenase (MADH), which became key enzymes for these species (*Anthony, 1982*; *Chistoserdova et al., 1991*). However, *Methylophilaceae* have been recently described that differ from these earlier characterized species with respect to their growth properties and the presence of MDH and/or MADH. For example, abundant unclassified marine *Methylophilaceae* represented by strain HTCC2181 demonstrate extremely slow growth and are inhibited by millimolar concentrations of C1 substrates (*Halsey, Carter & Giovannoni, 2012*). The genomes of these organisms encode neither MDH nor MADH (*Giovannoni et al., 2008*; *Huggett, Hayakawa & Rappé, 2012*). Similarly, *Methylotenera* species isolated from Lake Washington grow extremely poorly on methanol and possess no conventional MDH (*Kalyuzhnaya et al., 2006*; *Mustakhimov et al., 2013*). However, culture-independent experiments

indicated that these *Methylotenera* species are some of the major players in cycling of C1 compounds in the environment, as follows. In a previous study involving stable isotope probing (SIP) with $^{13}$C-labeled C1 compounds, *Methylotenera* sequences were enriched not only in communities accumulating heavy ($^{13}$C) carbon originating from methylamine, but also in communities accumulating heavy carbon from methanol or methane (*Kalyuzhnaya et al., 2008*). In a more recent SIP-based study we demonstrated that *Methylotenera* species, and specifically species devoid of genes for MDH synthesis were the major functional types, and they dominated over *Methylophilaceae* types possessing genes encoding proteins required for MDH as well as over other methylotroph species present in the site, suggesting a specific selection for MDH-negative genotypes (*Beck et al., 2013*). This study also suggested that *Methylotenera* species may be cooperating with methane oxidizers of the family *Methylococcaceae* in metabolizing methane, based on rapid accumulation of $^{13}$C label in the DNA of *Methylotenera* species (*Beck et al., 2013*). However, the metabolic nature of this cooperation remains unknown. The ratios of different ecotypes of *Methylophilaceae* were found different in response to different environmental conditions, suggesting distinctive ecological roles. In general, ecotypes most closely related to the previously described isolates *Methylotenera versatilis* 301 and *Methylotenera mobilis* JLW8 dominated the functional methylotrophic communities in these experiments (*Beck et al., 2013*). We have previously noted significant divergence between these strains when compared at the whole genome level, potentially reflective of specific adaptations of these strains or specific environmental functions (*Lapidus et al., 2011*).

The goal of this study was to obtain further insights into the metabolic potential of different *Methylophilaceae* and ultimately into their function in the environment, through comparative transcriptomics. We assess transcriptomes of three model *Methylophilaceae* under conditions approximating their natural environment and compare these transcriptomes to each other and to transcriptomes of cultures grown in laboratory conditions, in order to uncover whether different strains employ specific strategies for environmental adaptation. Two organisms, *M. versatilis* 301 and *M. mobilis* JLW8 were used as model organisms representing the most abundant ecotypes in our study site, Lake Washington sediment (*Beck et al., 2013*). We used *Methylovorus glucosotrophus* SIP3-4 as a representative *Methylophilaceae* strain containing MDH. This strain was found at low abundance at our chosen study site (*Beck et al., 2013*).

# MATERIALS AND METHODS

## Cultivation

*M. mobilis* JLW8 was cultivated in liquid mineral medium MM2 supplemented with 30 mM methylamine, as previously described (*Kalyuhznaya et al., 2009*; *Beck et al., 2011*). *M. versatilis* 301 was cultivated on plates as we were unable to cultivate this organism in liquid media (*Kalyuzhnaya et al., 2012*). Solidified diluted MM2 medium was used, supplemented with 30 mM methylamine, as previously described (*Kalyuzhnaya et al., 2012*). *M. glucosotrophus* SIP3-4 was cultivated in liquid medium using either methylamine

(30 mM) or methanol (100 mM) as substrates, as previously described (*Kalyuzhnaya et al., 2012*).

## Cultivation and sample preparation for transcriptomics experiments

The schematic of experimental design is depicted in Fig. 1. For transcriptomics (RNA-Seq) experiments, *M. mobilis* JLW8 and *M. glucosotrophus* SIP3-4 cells were grown on an appropriate substrate in liquid MM2 medium as described above to an $OD_{600}$ of approximately $0.45 \pm 0.05$. Stop solution (5% buffer-equilibrated [pH 7.4] phenol in ethanol) was added, and cells were harvested by centrifugation at $5000 \times g$ for 15 min at 4°C and immediately used for RNA extraction. *M. versatilis* 301 cells were grown on plates with solidified diluted MM2 medium supplemented with methylamine as described above. For methanol-induced samples, cells were grown on methylamine as described above, collected, centrifuged, washed with fresh diluted MM2 medium, and resuspended in MM2 plus methanol (25 mM, 50 ml total volume, in 250-ml flasks). After 2 h of incubation at 30°C, with shaking, cultures were pelleted as described above and immediately used for RNA extraction.

For *in situ* incubations, cells of *M. mobilis* JLW8 and *M. versatilis* 301 were grown on methylamine, and cells of *M. glucosotrophus* SIP3-4 were grown on methanol. Slide-A-Lyzer 3.5 K MWCO Dialysis Cassettes (12 ml Capacity, Thermo Fisher Scientific, Waltham, MA, USA) were used for sediment incubations, and these were prepared and cells were inoculated exactly as previously described (*Kalyuzhnaya et al., 2010*). Cassettes were placed on top of sediment cores that were collected as previously described (*Kalyuzhnaya, Lidstrom & Chistoserdova, 2004*), delivered to the laboratory on ice and used immediately. Cores were covered with foil and incubated for two days in a cold cabinet at 10°C. Cassettes were removed from sediment cores and cells were transferred into 50 ml tubes containing 0.5 ml of stop solution and collected by centrifugation at 5000 g for 15 min at 4°C. Two biological replicates were used for each condition.

## RNA extraction and ribosomal RNA depletion

RNA extraction was performed as described before (*Kalyuzhnaya et al., 2010*; *Beck et al., 2011*). The integrity of the RNA preparations was tested on a Bioanalyzer 2100 instrument (Agilent), using an Agilent RNA 6000 Nano kit as suggested by the manufacturer. The rRNA content was reduced using a MICROB*Express* Bacterial mRNA purification kit (Ambion). The RNA samples were submitted to a sequencing facility (the High-Throughput Genomics Unit, Department of Genome Sciences, University of Washington; http://www.htseq.org/index.html), where cDNA libraries were generated using a platform-specific (Illumina HiSeq 2000) chemistry. Sequencing was carried out on an Illumina HiSeq 2000 instrument using platform-specific protocols and producing reads of 36 bp in length.

### Data analysis

Reads corresponding to each sample were aligned to the respective reference genome (*Lapidus et al., 2011*) using the Burrows-Wheeler alignment tool (BWA; *Li & Durbin, 2009*) and using default parameters for small genomes. For each protein coding gene, the number

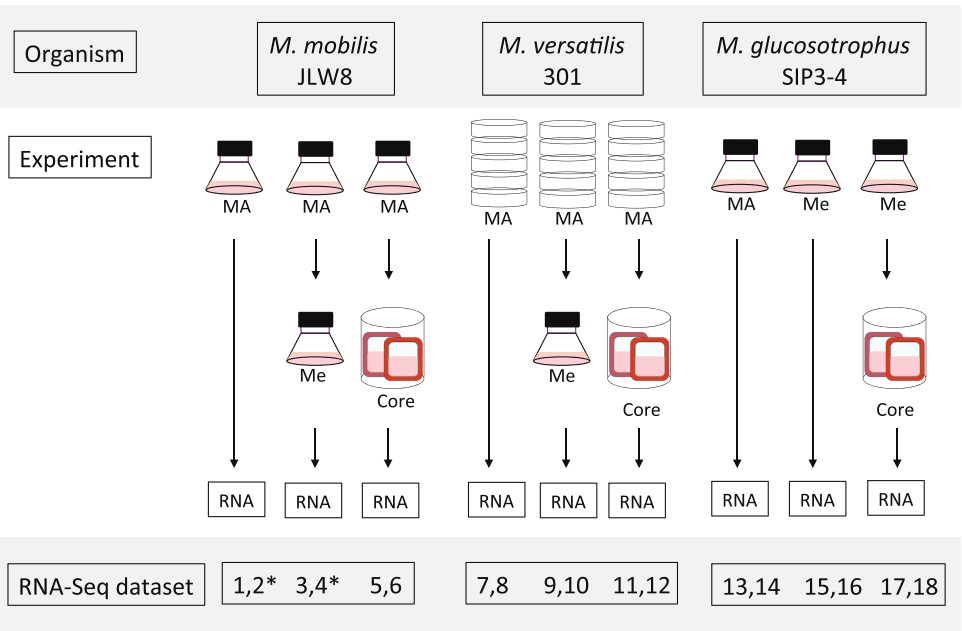

**Figure 1 Schematic of experimental setup.** *Methylotenera* species not possessing methanol dehydrogenase grow very poorly on methanol (*Kalyuzhnaya et al., 2006*; *Kalyuhznaya et al., 2009*). In addition, strain 301 cannot be cultivated in liquid culture (*Kalyuzhnaya et al., 2012*). We have previously developed protocols for specific gene induction and demonstrated that a rapid response to methanol typically occurs (*Beck et al., 2011*). Thus induction versus extremely long cultivation were chosen for the *Methylotenera* strains. *Methylovorus* species grow well on methanol (*Kalyuzhnaya et al., 2012*), and thus strain SIP3-4 was grown on methanol, not requiring induction experiments. Two biological replicate RNA samples were prepared, and two RNA-seq datasets were generated for each experiment (numbered 1 to 18). Datasets noted by asterisks have been previously described (*Beck et al., 2011*). Statistics for each dataset are shown in Table S1. MA, methylamine; Me, methanol.

of reads mapped per kilobase per one million reads (RPKM; *Mortazavi et al., 2008*) was computed. The RPKM data were averaged across biological replicates and visualized as an overlay on the genome using the tool described previously (*Hendrickson et al., 2010*). This visualization was used to identify genomic islands that were differentially abundant between conditions. Subsequently, the abundance measures of genes in the same pathway, function, or island as determined by expression pattern or prior knowledge, were summed and averaged across replicates to produce succinct plots comparing the three conditions for each organism.

## RESULTS

### Experimental setup and RNA-Seq statistics

Our main goal was to evaluate gene expression in the three model strains in semi-*in situ* conditions, i.e., after incubation in semi-penetrable dialysis cassettes placed into freshly-sampled lake sediment cores, assuming that metabolic processes that naturally occur in this environmental niche continued over the short duration of time (48 h) in the laboratory. Note that control samples, i.e., samples growing under laboratory conditions,

were prepared differently for each strain (Fig. 1). Cells of *M. glucosotrophus* SIP3-4 were grown on either methanol or methylamine to exponential phase and harvested for RNA isolation. Cells of *M. versatilis* 301 were grown on methylamine-supplemented plates and either harvested for RNA isolation or harvested and incubated with methanol before RNA isolation. Cells of *M. mobilis* JLW8 were grown in liquid cultures on methylamine and either harvested for RNA isolation or incubated with methanol as previously described (*Beck et al., 2011*). Obviously, the different growth conditions necessitated by the specific physiology of the strains employed may have affected gene expression patterns. Thus we compare expression patterns for different cultures cautiously. However, for each strain, comparing the *in situ*-incubated cultures to two laboratory culture controls (methanol and methylamine) should provide good clues as to which transcriptional responses are specific to *in situ* conditions. A total of 18 samples were compared in this study (Fig. 1), of which 14 samples were generated in this study and 4 were generated previously (*Beck et al., 2011*). Sequencing and transcript quantification statistics are shown in Table S1. In each transcriptome, matches were found for 98.0 to 100% of the annotated genes. For each gene, RPKM was used as the measure of abundance. Close correlation was observed between biological replicates for each experimental condition, with the correlation coefficient between replicates ranging from 0.84 to 0.99 (Table S1). The actual RPKM data for the newly generated datasets along with fold change and statistical analyses are shown in Tables S2–S10. A global view of the condition-condition comparisons for each organism as shown in Fig. 2 indicates that most of the genes were expressed at similar levels in each condition, suggesting that differential expression of certain genes was in response to the specific conditions rather than to sample-to-sample noise, or as a general reflection of stress. On another hand, differential expression for some of the housekeeping genes was predicted. For example, the ribosomal protein synthesis gene cluster was less expressed in core samples compared to laboratory samples, as a reflection of slower metabolism (not shown), a trend noted in our previous study (*Kalyuzhnaya et al., 2010*).

We further focused our attention on two main groups of genes: the genes known for their role in methylotrophy, in order to investigate their contribution to *in situ* metabolism, and genes that demonstrated the most pronounced transcriptional response to the *in situ* conditions, in order to delineate candidate pathways/functions as important for environmental adaptation. Genes involved in a specific metabolic pathway/function were considered, in most cases, in a modular fashion, by summing RPKM counts for the genes involved. The metabolic pathways, specifically the methylotrophy pathways or other previously annotated pathways were considered based on prior knowledge (*Chistoserdova, 2011a*; *Chistoserdova, 2011b*) and not always included contiguous genes on the chromosomes. These pathways are outlined in Fig. 3 with genes belonging to each specific metabolic module listed in Table S11. Genes for the functions not previously studied in this group of organisms were determined by their contiguous location on the chromosomes and by their expression pattern.

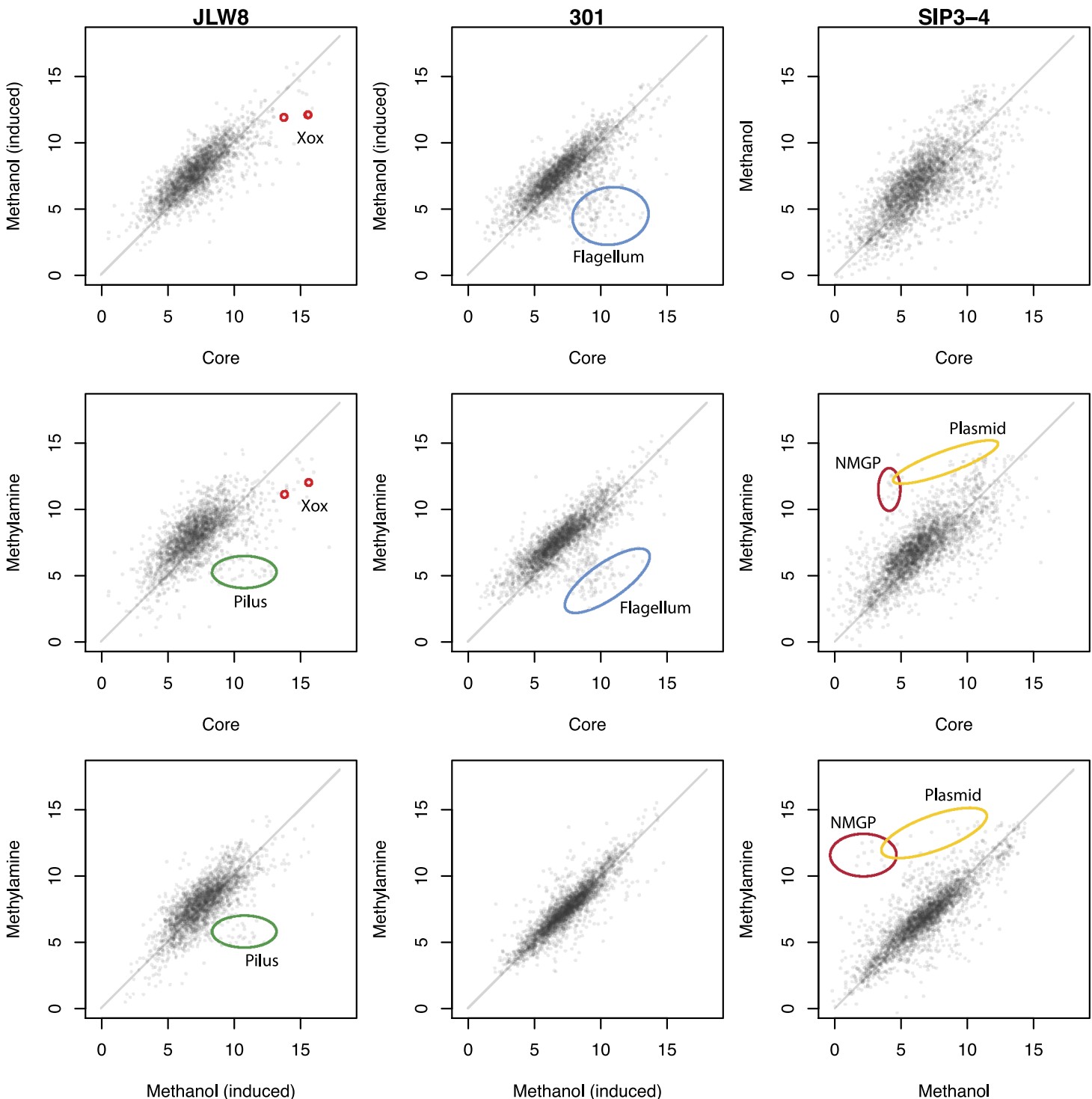

**Figure 2 A global view of transcriptome/transcriptome comparisons for each organism.** Each point is the mean (across replicates) log2 RPKM for one condition vs. another, as indicated. Colored circles highlight some of the differentially expressed modules. Pilus, the unique Type II secretion module; Xox, the *Mmol_1769/Mmol_1770* gene pair; Flagellum, flagellum cluster 1; NMGP, genes of N-methylglutamate pathway for methylamine oxidation.
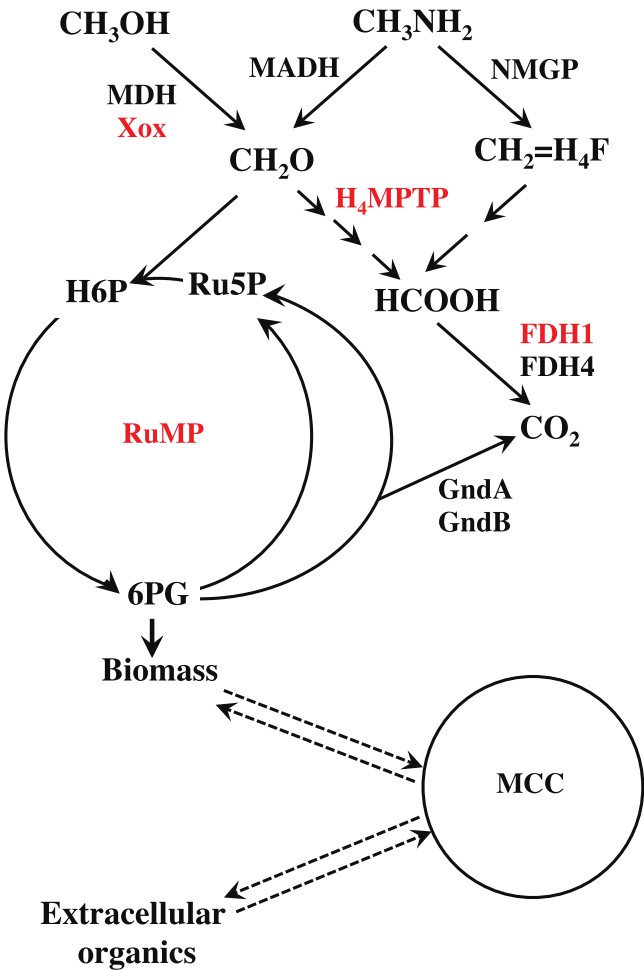

**Figure 3 Schematic of central metabolism of *Methylophilaceae* and major metabolic modules addressed in this study.** As the role of MCC remains unknown, dashed lines indicate potential scenarios for its intermediates originating from central metabolic pathways or from external sources. Modules common to all three organisms indicated in red. Modules not shared by all three organisms are shown in black. See Table S11 for details. MDH, methanol dehydrogenase; MADH, methylamine dehydrogenase; NMGP, N-methylglutamate pathway; H4MPT, tetrahydromethanopterin-linked pathway for formaldehyde oxidation; RuMP, ribolemonophosphate pathway; FDH1,2, formate dehydrogenase enzymes; GndA,B, 6-phosphagluconate dehydrogenase enzymes; MCC, methylcitric acid cycle.

## Expression of *xoxF* genes suggests different regulation and different roles for multiple homologs

We have previously noted that multiple homologs of *xoxF* genes (that are all homologs of the large subunit of methanol dehydrogenase; *Chistoserdova, 2011b*) were present in the genomes of the three organisms, and more recently we have reported on the phenotypes of *xoxF* mutants in *M. mobilis* JLW8 that suggested these genes must encode enzymes involved in methanol oxidation, even though methanol dehydrogenase activity could not be measured in this organism (*Mustakhimov et al., 2013*). Thus the expression of these genes was of special interest. We demonstrate elevated expression of *mmol_1770* in *M. mobilis* JLW8 in response to the *in situ* conditions (4 to 6-fold compared to

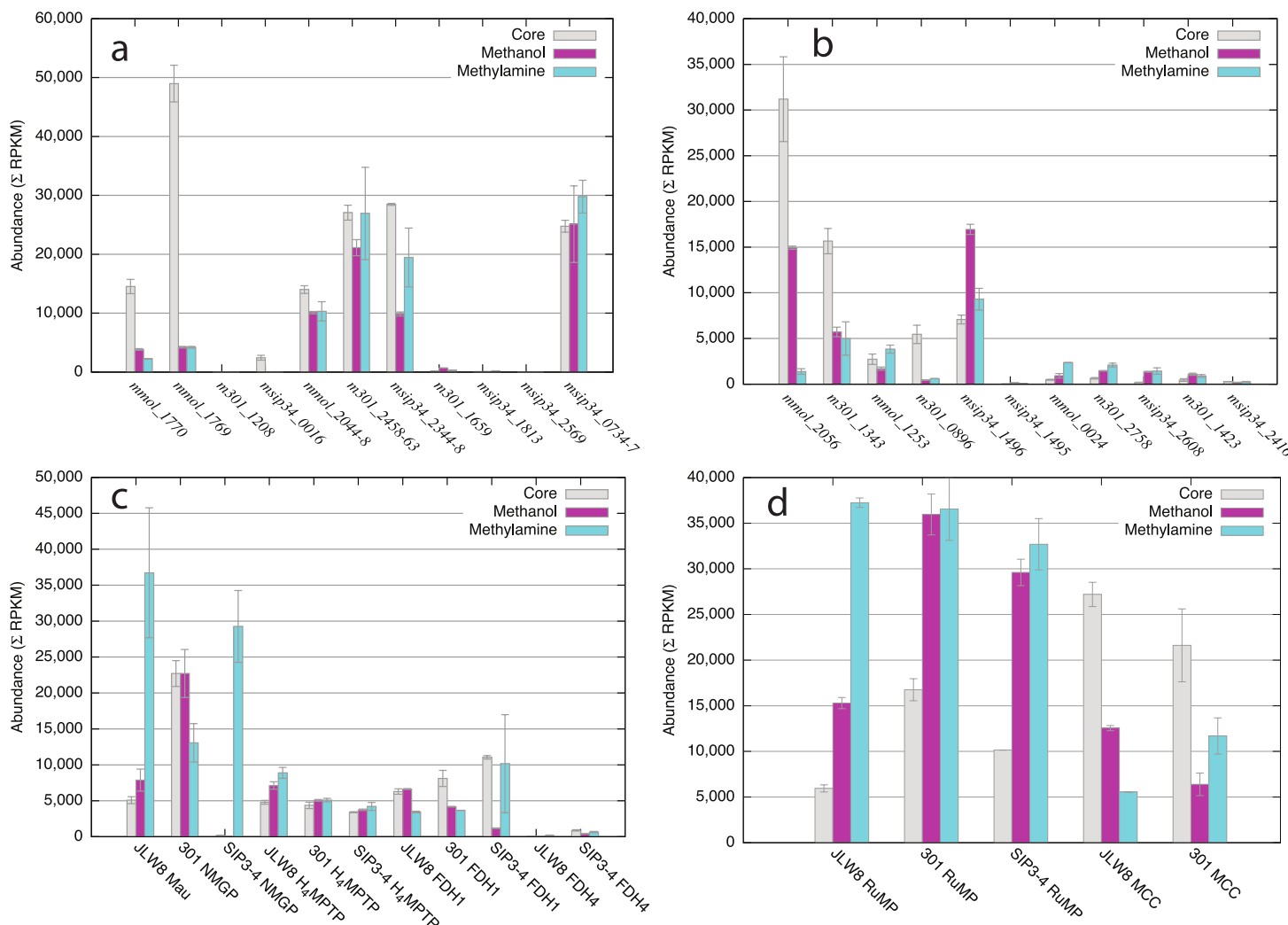

**Figure 4** **Relative abundance of transcripts reflecting expression of central pathways for carbon assimilation and dissimilation.** (A) Abundance of transcripts from different homologs of xoxF genes and genes implicated in encoding other xox functions, compared to abundance of transcripts matched to the mxaFJGI cluster encoding methanol dehydrogenase. (B) Relative abundance of transcripts from different homologs of fae or fae-like genes. (C) Combined relative abundances of transcripts from genes encoding major methylotrophy pathways. Mau, methylamine dehydrogenase (*mmol_1567-1576*); NMGP, N-methylglutamate pathway (*m301_1414-1421* and *msip34_2421-2428*); H4MPTP, H4MPT-linked formaldehyde oxidation pathway (*mmol_0858-0862, 0896-0899, 1331-1336, 1342-1347; m301_0909-0913, 0950-0953, 1540-1545, 1552-1557; msip34_1489-1494, 1500-1509, 1683-1685*); FDH1, FDH4, non-homologous formate dehydrogenases (*mmol_2031-2035* and *mmol_0469, 0470; m301_2445-2449; msip34_1177-1181* and *msip34_1599, 1600*). (D) Combined relative abundances of transcripts from genes for the methylcitric acid cycle (MCC; *mmol_0748-0766; m301_0686-0703*), compared to abundances of transcripts for the ribulose monophosphate pathway (RuMP; *mmol_0287, 0313, 1337-1339, 0827, 1429, 1526, 1527, 1726, 1727, 1980, 2239; m301_0182, 0304, 1112, 1114, 1501, 1546-1548, 1566, 2018, 2019, 2400, 2582; msip34_0164, 0268, 0483, 1093-1095, 1138, 1269, 1497-1499, 1896, 1897, 2516*).

laboratory cultures (Fig. 4A, also highlighted in Fig. 2). Along with the well-characterized methylotrophy genes (e.g., *fae* encoding formaldehyde-activating enzyme, a key enzyme in dissimilatory metabolism; *hps* encoding hexulosephosphate synthase, a key enzyme in assimilatory metabolism; *Chistoserdova, 2011b*), *mmol_1770* was one of the most highly expressed genes in the *in situ* conditions in this organism (Tables S2–S4). A neighboring gene *mmol_1769* predicted to encode a small cytochrome (not homologous to *mxaG*

encoding a specific cytochrome for MDH; *Chistoserdova & Lidstrom, 2013*) was also highly expressed (Fig. 4A), suggesting a role for this cytochrome as a specific electron acceptor from *Mmol_1770*. In contrast, in *M. versatilis* 301, a homolog of *mmol_1770* (*m301_1208*) was expressed at a very low level in each condition, and no homolog of *mmol_1769* was present near *m301_1208* or elsewhere in the genome, suggesting no role for this gene in either methylotrophy or in environmental niche adaptation for this organism (Fig. 4A, Tables S5–S7). The homolog of *mmol_1770* in *M. glucosotrophus* SIP3-4 (*msip34_0016*) was significantly (100-fold) over expressed in *in situ* conditions compared to laboratory conditions, but the expression level was relatively low (Figs. 4A, Tables S8–S10). The neighboring gene (*msip34_0015*) predicted to encode a cytochrome (not homologous to either *mmol_1769* or *mxaG*) revealed a similar expression pattern, suggesting a role in electron transfer from Msip34_0016.

The second *xoxF* homolog in *M. mobilis* JLW8, *mmol_2048* was also slightly over-expressed in *in situ* conditions. The corresponding genes in *M. versatilis* 301 and *M. glucosotrophus* SIP3-4 were also most highly expressed *in situ*, compared to the methylamine- and methanol-grown or induced cultures. Figure 4A shows sums of abundances of reads matching these genes and what appear to be accessory genes *xoxJ* and *xoxG*, along with an oxidoreductase of unknown function as these genes form tight and highly conserved clusters in all known *Methylophilaceae* genomes (*Lapidus et al., 2011*).

One additional *xoxF* homolog in *M. versatilis* 301 (*m301_1659*) and two additional homologs in *M. glucosotrophus* SIP3-4 (*msip34_1813* and *msip34_2549*) were expressed at low levels. Of the three organisms, only *M. glucosotrophus* SIP3-4 encoded the well-characterized (MxaFI) methanol dehydrogenase (*Chistoserdova, 2011b*), and the genes encoding this enzyme, including accessory genes (*msip34_0734-0737*) were highly expressed on all substrates as has been previously shown for other methylotrophs (*Okubo et al., 2007*; *Hendrickson et al., 2010*).

## Differential expression of *fae* genes further supports dual functionality

While one function of formaldehyde activating enzyme (Fae), in linking formaldehyde to tetrahydromethanopterin ($H_4MPT$), is well characterized (*Vorholt et al., 2000*), we have previously proposed that Fae enzymes may have additional functions, for example in sensing, signaling or regulation (*Kalyuzhnaya et al., 2008*). Thus we analyzed expression of *fae* and *fae*-like genes separately from other genes involved in the $H_4MPT$-linked pathway for formaldehyde oxidation. We found that the *fae* subtype that has been previously determined as a novel subtype in *M. mobilis* based on phylogenetic analysis (*Kalyuzhnaya et al., 2008*) was overexpressed in sediment samples in both *M. mobilis* JLW8 and *M. versatilis* 301, while this type was not encoded by *M. glucosotrophus* SIP3-4 (Fig. 4B). This gene does not appear to be part of a gene cluster in either of the organisms, and the adjacent genes are not conserved. The conserved subtypes of *fae* (i.e., the subtypes present in all three organisms) showed different transcription patterns, as follows. *mmol_1253* that is part of a sensing/chemotaxis gene cluster (*Lapidus et al., 2011*) was transcribed at a much lower level compared to *mmol_2056* and revealed no response to *in situ* conditions while

*m301_0896* was approximately 10-fold over expressed in the sediment samples compared to methanol or methylamine (Fig. 4B). The surroundings of these two genes are only partially conserved (*Lapidus et al., 2011*). *msip34_1496* that is part of the H₄MPT-pathway gene cluster was most highly expressed on methanol, while the neighboring homolog *msip34_1495* was transcribed at a very low level. The *fae2* and *fae3* types of *fae* homologs were transcribed at moderate levels and showed no response to environmental conditions.

## Expression of other oxidative methylotrophy metabolic modules supports their proposed functions while highlighting a few variations

*M. mobilis* JLW8 uses methylamine dehydrogenase to metabolize methylamine (*Kalyuzhnaya et al., 2006*; *Lapidus et al., 2011*). The 10 *mau* genes were all expressed at much higher levels in methylamine cultures compared to methanol-induced cultures, and the expression was even lower in the core conditions (Fig. 4C) suggesting that this gene cluster responds to availability of methylamine as is shown for other organisms utilizing Mau systems (*Chistoserdov et al., 1994*; *Delorme et al., 1997*). *M. versatilis* 301 and *M. glucosotrophus* SIP3-4 utilize an alternative metabolic pathway to oxidize methylamine, the N-methylglutamate pathway (*Latypova et al., 2010*; *Kalyuzhnaya et al., 2012*). While in *M. glucosotrophus* SIP3-4 the genes for this pathway, encoding subunits of N-methylglutamate dehydrogenase, $\gamma$-glutamylmethylamide synthase and N-methylglutamate synthase (Table S11), were only expressed in methylamine-grown cells, in *M. versatilis* 301 they were expressed in all conditions, suggesting either different regulatory mechanisms for this pathway in different organisms or reflecting the fact that repression of this pathway did not occur over the timescale of the experiment. Expression of the genes of the H₄MPT-linked pathway (20 genes, excluding the *fae* or *fae*-like genes) was not greatly influenced by the conditions, and the levels of expression were similar for the three organisms. All three organisms expressed genes for a NAD-linked formate dehydrogenase known as FDH1 (*Hendrickson et al., 2010*; *Lapidus et al., 2011*) while genes for an alternative enzyme known as FDH4 (*Hendrickson et al., 2010*; not encoded by *M. versatilis* 301, *Lapidus et al., 2011*) were expressed at low levels (Fig. 4C).

## Expression pattern of the methylcitric acid cycle further suggests a role in environmental adaptation of *Methylotenera*

Some *Methylophilaceae* species (in this study, *Methylotenera* but not *Methylovorus*) possess genes encoding the methylcitric acid cycle (MCC), whose role remains unknown in these species (*Chistoserdova, 2011a*). We found that in both *Methylotenera* species, the genes for MCC were highly expressed in *in situ* conditions (Fig. 4D), in general higher than the genes of the ribulose monophosphate pathway (RuMP), the main carbon assimilatory pathway in *Methylophilaceae* (*Chistoserdova, 2011b*; *Chistoserdova & Lidstrom, 2013*). One of the most over expressed genes in the MCC gene cluster in both organisms was a *gntR*-type gene predicted to encode a transcriptional regulator (*mmol_0754* and *m301_0692*), suggesting a role for this gene in switching the pathway on in response to environmental conditions. Transcription of the RuMP genes followed a reversed pattern in these organisms, being

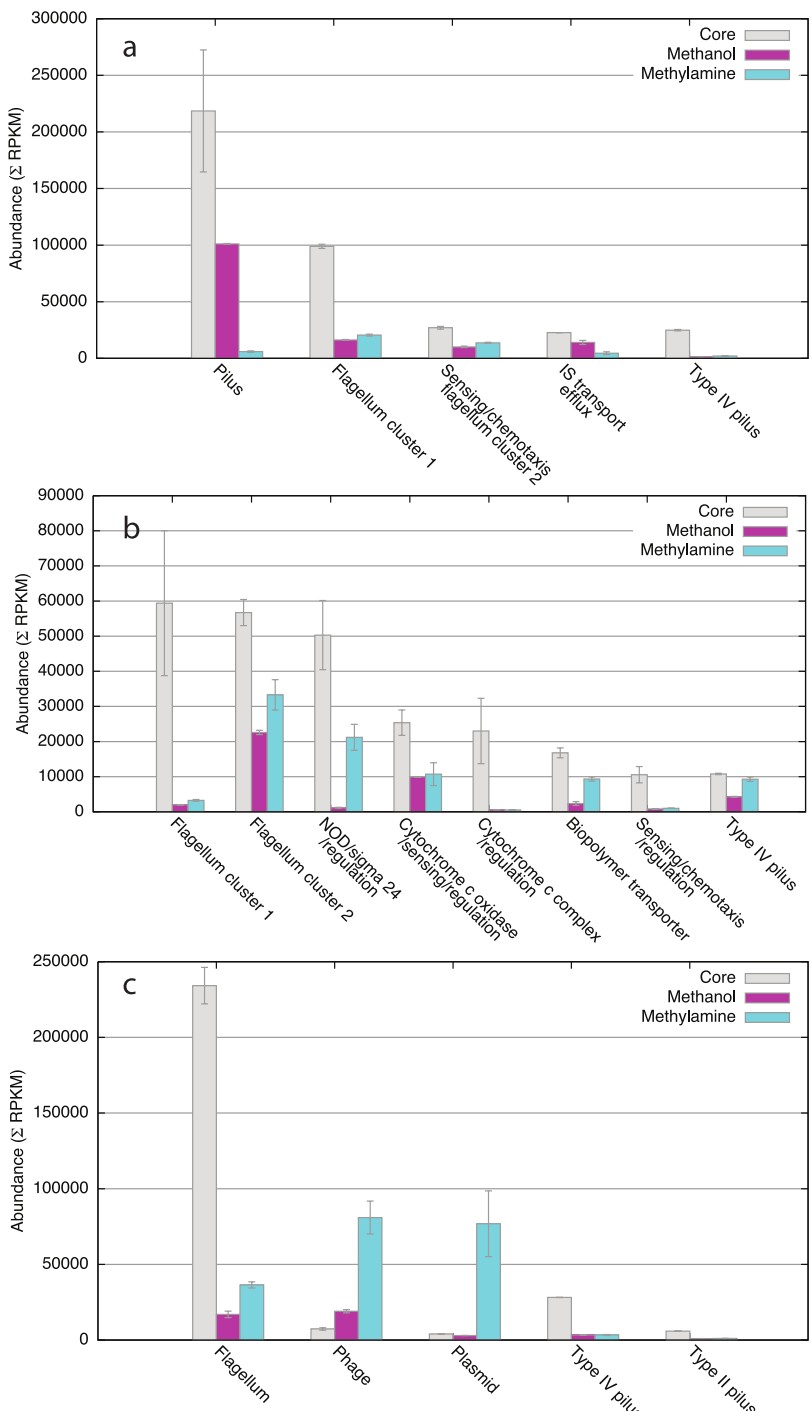

**Figure 5 Combined relative abundances of transcripts for genes most highly and most differentially expressed in each of the three organisms.** (A) *M. mobilis* JLW8 unique pilus genes, *mmol_1380-1409*; flagellum cluster 1, *mmol_0919-0953*; sensing/chemotaxis/flagellum cluster 2, *mmol_1248-1273*; IS transport/efflux, *mmol_1451-1493*; Type IV pilus *mmol_2269-2274*. (B) *M. versatilis* 301 flagellum cluster 1, *m301_1679-1716*; flagellum cluster 2, *m301_0961-1015*; (continued on next page...)
highly expressed in laboratory conditions (on both methylamine and methanol) and transcribed at lower levels *in situ*.

## Motility/adhesion functions appear of special importance in *in situ* function

When we considered genes that were both highly transcribed and over expressed in the *in situ* conditions, genes with such a pattern mostly represented the following major functional categories: motility, adhesion, signaling and sensing, and transport. The gene cluster(s) encoding the flagellum functions, including regulatory functions, were some of the most highly transcribed genes in all three organisms (Figs. 5A–5C), many of these genes being conserved among the three organisms. One other conserved function over expressed in all species was a Type IV secretion function encoded by a cluster of 6 genes (*mmol_2269-2274*, Fig. 5A; *m301_2616-2621*, Fig. 5B; *msip34_2593-2598*, Fig. 5C, encoding homologs of PilE, PilV, PilW, PilY, FimT and a hypothetical protein). In *M. glucosotrophus* SIP3-4, a Type II secretion gene cluster was also over expressed (*msip34_2502-2507*; approximately 6 fold) in *in situ* conditions (Fig. 5C). This gene cluster is strongly conserved in other organisms, and the respective genes are all transcribed in the *Methylotenera* species. However, in these latter species, no response to *in situ* conditions was observed, suggesting that the same type of a functional system may have a different response and be regulated differently *in situ*.

## Unique genomic elements of each organism demonstrate differential abundance

Remarkably, most of the other significantly over expressed functions were unique to each organism. In *M. mobilis* JLW8, one of the most highly expressed functions was the gene cluster encoding a Type II secretion system that is part of a genomic island previously identified based on genome/genome comparisons and analysis of GC value deviation (*Lapidus et al., 2011*). We previously noted that this gene cluster was dramatically overexpressed in response to exposure to methanol, compared to growth on methylamine (*Beck et al., 2011*). From the analyses conducted here, the expression of this gene cluster *in situ* is similar to the expression induced by methanol, while the expression is dramatically down during growth on methylamine (up to 1000-fold dependent on the gene considered; Fig. 5A). Other gene clusters that were uniquely present and responding to *in situ* conditions at the transcriptional level included an insertion sequence (IS) element (*mmol_1456-1493*) encoding transport/efflux functions in *M. mobilis* JLW8 (Fig. 5A), and predicted biopolymer transport functions (*m301_2524-2533*) in *M. versatilis* 301 (Fig. 5B). This latter gene cluster was partially conserved in the two other organisms, but expression
of the counterpart genes was not significantly changed *in situ* compared to laboratory growth.

Over expressed functions unique to *M. versatilis* 301 included a gene cluster encoding (FAD-binding flavohemoglobin) nitric oxide dioxygenase, a sigma 24 factor and a transcriptional regulator (*m301_0769-0776*), a cytochrome complex and a two-component transcriptional regulation system (*m301_0064-0069*), a cytochrome oxidase along with sensing and regulation functions (*m301_844-858*), and a sensing/chemotaxis/regulation cluster (*m301_0896-0902*; Fig. 5B).

In *M. glucosotrophus* SIP3-4, a few prominently transcribed functions were found to be significantly repressed *in situ* compared to laboratory conditions, most notably a phage-like genomic island (*msip34_2008-2089*) previously identified by observations of a dramatically different GC content (*Lapidus et al., 2011*) and one of the two plasmids (*msip34_2831-2843*; Fig. 5C). The roles of these elements in physiology of *M. glucosotrophus* SIP3-4 remain unknown.

## DISCUSSION

This work is an important step toward a better understanding of the physiology of different *Methylophilaceae* species in their natural niche, lake sediment, where they likely play an important role in the cycling of carbon and nitrogen, as part of a diverse methylotroph community. We have previously shown that *Methylophilaceae* and most prominently the *Methylotenera* species not possessing MDH respond to methane, methanol and methylamine stimuli (*Kalyuzhnaya et al., 2008*; *Kalyuhznaya et al., 2009*) and that they are likely engaged in a cooperative behavior with *Methylococcaceae* species (*Beck et al., 2013*). However, little knowledge exists on what metabolic pathways or what other functions are required for these organisms to perform their environmental role. Glimpses into the environmental function of *M. mobilis* JLW8 have been gained previously using an *in situ* setup similar to the one described here, in combination with environmental array-based transcriptomics (*Kalyuzhnaya et al., 2010*). Here we expand the transcriptomics approach by including additional, genetically divergent *Methylophilaceae* species and by employing deep sequencing-based transcriptomics that provides semi-quantitative access to the whole transcriptomes, and thus significantly better resolution is achieved through this approach. We demonstrate that the major mechanisms for environmental adaptation include both methylotrophy and non-methylotrophy metabolic modules as well as modules involved in other functions. Our data also demonstrate that, besides shared strategies, the organisms employed in this study also utilize strategies unique to each species, suggesting that the genomic divergence plays a role in environmental function and niche fitness. At the same time, it appears that homologous gene/protein systems may play different roles in different organisms.

One of the notable outcomes of this study is further evidence for the importance of XoxF enzymes in the metabolism of the *Methylophilaceae in situ*. However, it appears that different species have harnessed different homologous Xox systems as major metabolic modules. While in *M. mobilis* JLW8 the homolog associated with a small, unique

cytochrome appears to be the major module responding to environmental conditions, other species investigated here appear to rely on the alternative XoxF systems. As the only MDH-containing species included in this study, *M. glucosotrophus* SIP3-4 appeared to have decreased transcription from the *mxaFJGI* gene cluster in response to *in situ* conditions and an increased transcription from two of the four *xoxF* (and accessory) genes.

Different homologs of *fae* also demonstrated different responses to *in situ* conditions, further suggesting that different homologs may respond to different signals, and, in turn, that different homologs may have different additional functions, as proposed earlier (*Kalyuzhnaya et al., 2008*).

This study reinforces the prior observations on the apparent importance of the methylcitric acid cycle (MCC) in the *Methylotenera* species. High transcription from the respective gene clusters and pronounced over-expression of the genes in the pathway, along with the reverse expression pattern for the ribulose monophosphate pathway genes suggest a switch in central metabolism of *Methylotenera* in response to *in situ* conditions. The nature of the metabolite(s) processed via this pathway still remains unknown.

Of the functions not related to central metabolism, flagellum and fimbria synthesis functions appeared to be of significance for environmental function, based on their high abundance and differential expression. Obviously, solely from gene/protein homology it is impossible to predict the major functions of the flagella or of different pili in different *Methylophilaceae* strains, the former being implicated in not only mobility but also attachment and adhesion (*Römling, Galperin & Gomelsky, 2013*), and the latter implicated in a variety of functions, including adhesion, protein transfer, virulence, conjugation and transformation, to mention a few (*Zechner, Lang & Schildbach, 2012*; *Giltner, Nguyen & Burrows, 2012*; *Campos et al., 2013*). Finding differential expression for different types of pili in different species, while intriguing, is so far not sufficient for proposing specific strategies of communal functioning. Instead, at this time, these are only suggestive of prominent targets for further experimental investigations, via genetic manipulations of single species/communities and via expanded-omics approaches.

### Funding

L Chistoserdova and ME Lidstrom received grants from the National Science Foundation (MCB-0950183 and MCB-0604269). This research was facilitated through the use of advanced computational, storage, and networking infrastructure provided by the Hyak supercomputer system, supported in part by the University of Washington eScience Institute. The funders had no role in study design, data collection and analysis, decision to publish, or preparation of the manuscript.

### Grant Disclosures

The following grant information was disclosed by the authors:
National Science Foundation: MCB-0950183, MCB-0604269.

## Competing Interests

L Chistoserdova is an Academic Editor for PeerJ.

## Author Contributions

- Alexey Vorobev performed the experiments, wrote the paper.
- David A.C. Beck analyzed the data, wrote the paper.
- Marina G. Kalyuzhnaya conceived and designed the experiments, performed the experiments, wrote the paper.
- Mary E. Lidstrom conceived and designed the experiments, wrote the paper.
- Ludmila Chistoserdova conceived and designed the experiments, analyzed the data, wrote the paper.

## Supplemental Information

Supplemental information for this article can be found online at http://dx.doi.org/10.7717/peerj.115.

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
