# Peer review of "Comparative transcriptomics in three Methylophilaceae species uncover different strategies for environmental adaptation"

_PeerJ, doi:10.7717/peerj.115_

## Round 0.1 · original submission · Minor Revisions

As you can see, all three reviewers suggest that the paper needs to be revised before it can be accepted for publication. Therefore, I would be grateful if you would consider and address all of their comments in revising your paper. In addition, please would you address the points below.

1. Extend and broaden the Abstract (only 124 words as submitted) to (a) highlight the experimental approach (e.g. the use “focused transcriptomics” and a limited range of growth conditions related to methylotrophy); and (b) make the significance of the results clearer to readers who are not so familiar with C1 metabolism and methylotroph ecotypes.

2. Modify the presentation of the figures to: (a) make the legends more comprehensive so that the figures are understandable and self-contained, in isolation from the text (e.g. provide more information about the sample treatments/cultivation conditions, rather than the highly abbreviated axes labels (Fig. 1) and keys (Figs 2-8); and (b) combine some of the figures as suggested by Reviewer 2 – I would suggest Figs 2-5 and Figs 6-8 should be combined in this way. Please specify “sediment core”, here and in the main text.

3. Specify the DNA sequencing platforms used in this study (p. 6, lines 4-6).

4. Generally make the paper easier for “non-C1 afficionados” to read. I like Reviewer 2’s suggestion to include a figure illustrating the methylotrophic pathways found in the Methyophilaceae, annotated to include gene designations linked to the main results.

5. Check MS carefully for typos and inconsistencies in reference citations (e.g. Baev et al 1991/1992 [p.2, line 15], Lapidus 2001/2011 [p. 11, line 4], Lidstrom et al [p. 22, lines 14-16] not cited in text).

·

Basic reporting

See below

Experimental design

See below

Validity of the findings

See below

Additional comments

The manuscript by Vorobev et al. is a comparative study describing the responses of three Methylophilaceae species to C-1 compounds and how this could relates to previous environmental observations on this microbial group. This is a transcriptomics study and as expected the results were complex. The authors did a nice job of presenting the results, focusing on the changes in C-1 metabolism. They also did a nice job noting the unexpected results only briefly. Thus, the manuscript should be considered for publication after addressing the minor points listed below.

Specific Comments

1. Page 4 lines 15 and 16: change “does not grow in liquid media” to “unable to cultivate on liquid media.” The authors need to demonstrate the organism requires a solid surface for growth to make this claim. The more likely explanation is some growth factor is supplied in the agar or the initial C-1 concentration in the liquid medium is too high (i.e. C-1 concentration adjacent to colony is lower), a slow C-1 feed fermentor/chemostat may solve this issue .
2. Page 6 line 11: define RPKM on first usage

·

Basic reporting

see report

Experimental design

see report

Validity of the findings

see report

Additional comments

This is a nice paper which uses transcriptomics to start to look at the responses of key methylotrophic species to different environmental growth conditions. The paper is generally well written and the experimental design and methodology is sound. The growth conditions used are fairly narrow but I like the use of the dialysis cassette to try to mimic environmental conditions more like those in situ. More could be made of this method as a way to get closer to natural environmental conditions. I also like the “focussed transcriptomics” experiments because the authors are attempting to link changes in expression of key genes involved in a specialised metabolism, rather than trying to analyse everything. The results obtained also strengthen the importance/significance of XoxF and the MCC pathway in methylotrophs.
I think it would strengthen the readablilty of the manuscript if a figure (such as that found in the authors ASM review) of methylotrophic pathways found in the Methyophilaceae was added. It would also be useful to have a table of all of the methylotrophy/methylotrophy related genes being analysed, together with their function (or putative function). Again this would aid reading for the non-C1 afficionados. Some of the subsequent figures (2-7) could be grouped together.
Otherwise I have only minor comments which I hope will aid the flow/clarity of the manuscript.
Page 2
line 9. A few lines to expand on Giovannoni et al and unpublished observations is warranted here.
Line 14 please define C1 for the generalist reader.
Line 20 hallmark enzymes? Key enzymes?
Page 3
Line 3 traditional = conventional?
Line 8 heavy (13C-labelled) carbon
Line 11 genes encoding proteins required for MDH synthesis
Line 15 Rephrase engaged in cooperation please
Line 17 I suggest: on rapid accumulation of 13C label in the DNA of….. PLUS, this could do with a few more lines of explanation.
Line 19 might be better as The ratios of different ecotypes of Methylophilaceae could… (avoid dynamics)
Line 22 functional methylotrophic communities….
Page 4
Line 1 might be better: ultimately into their function in the environment.
Line 2
Of three model Methylophilaceae under conditions approximating their…
Line 5
The two methylotrophs M. versatilis 301 and M. mobilis JLW8 were used as model organisms representing….
Line 8
We used Methylovorus glucosotrophus SIP3-4 as a representative (model) Methylophilaceae strain containing MDH and which was present at low abundance at our chosen study site.
Page 5 The use of the dialysis cassette is neat-I think the experimental set-up warrants a figure.
Page 7
Line 2 give timescale please
Page 8
Line 24
mmol_1770. Is this xoxF? A table would be useful giving enzyme/gene /ORF
Page 9
Line 2-say what fae is and that HMP synthase is key enzyme in C assimilation etc (or use a table)
Line 5 is this a cytochrome CL like cytochrome?
Line 17 counterparts = corresponding genes
Line 19 please rephrase followed by methanol and methylamine to be more precise.
Line 25 true= maybe better to say functional? Ie enzyme activity ascribed to these (genes) proteins.
Page 10
Line 8
sensing of what? Formaldehyde?
Line 14-15
its surroundings = genome context ie adjacent genes
line 15 phylotypes? Please explain

Page 11
Line 11
State what these genes are
Page 12
genes encode enzymes of the MCC pathway-please revise
Line 5
the main carbon assimilatory pathway
Line 7
Presumably gntR is a transcriptional activator? Please state likely function/known “homologs etc
Line 11 in laboratory conditions (with which substrates?)
Line 20 Type IV secretion function: what are likely proteins/homologs?
Line 24
This gene cluster….

Page 13
Line 11 overexpressed during growth on methanol compared to growth on…
Line 13 …
… during growth on methylamine
Line 22
Nitric oxide dioxygenase is this still considered a flavohemoglobin? What type of sigma factor is it? What are the regulatory functions? Transcription factors? A few more line of detail would be useful here.
Line 14
Line 2 phage like genomic island. Is this on a plasmid?
Page 15
Line 25-26. Could the flagellum and fimbria synthesis be due to the energy metabolism/status of the cells (lab growth versus semi-in situ growth conditions?)

Reviewer 3 ·

Basic reporting

This article is well written and self-explained. A comparative transcriptomics approach (RNA-seq) is used in order to dissect metabolic pathways used by the three selected beta-proteobacterial methylotrophs under different cultivation strategies.

Experimental design

Experimental design is clear, and in general fits well with the aim of the study. It may however benefit by clarifying the following points
1. It is unclear to me why cultivating Methylotenera first on methylamine then inducing them with methanol, rather than cultivating directly on methanol. Is it because they grow poorly on methanol? Please clarify.
2. I had assumed it's a common practise to validate direct comparison of RNA-seq by qRT-PCR, particularly for the genes of interest, e.g. xoxF, fae, genes encoding methylcitric acid cycle and H4MPT pathway.
3. RPKM data for some of those commonly used "house-keeping" genes seem to vary up to a few folds. Is it worth a mention in the manuscript?
3. I appreciate that RNA-seq can be expensive, but the authors may want to justify the use of 2 biological replicates. As the authors mentioned that the focus of this study was to “uncover whether different strains employ specific strategies for environmental adaption”. One would think a direct comparison of laboratory methylamine culture to “near in situ” incubation is more desirable given that methanol is probably not a best substrate for the selected strains anyway. This would generate the same amount of samples but allow triplicate for each.

Validity of the findings

no comments

Additional comments

RNA-seq generates tremendous amount of data and It is inevitable that the authors have cherry-picked a few metabolic pathways of their interest. However from the readers’ point of view there are a number of other genes which have been massively up-regulated in "near in situ" conditions and many of them are annotated as hypothetical proteins. Because the authors have the particular expertise in the field, I wonder if they have any insight on the putative functions of those genes or indeed whether they have planned to investigate the functions of these hypothetical proteins?

---

## Round 0.2 · accepted · Accept

The revision addresses all of the substantive points raised in review, and incorporates almost all of the amendments suggested by the reviewers. However, there are a few further points, listed below, that I would ask you to attend to prior to publication.

1. Strangely, apart from section breaks, the main text contains no paragraphs, which makes the paper more difficult to read. I would suggest paragraphs as follows, but I am very happy for you to make amendments to improve the readability of the paper: p2, l11, l22; p3, l4, l22; p4, l3; p5, l13;p7, l7, l18; p8, l3, l11; p9, l11, l24; p10, l3, l10; p12, l6, l25; p14, l6, l12; p15, l16, l23; p16, l9, l15.

2. Reviewer 2 questioned use of the term phylotype with respect to fae (formaldehyde activating enzyme) genes (now on p11, l4-5). Furthermore, I am unsure how you distinguish between phylotype (your undefined term), type and homolog (e.g. on p11, l12-14). I think your use of the term phylotype (normally reserved for a clade of organisms related to each other by a defined genetic distance) in this context is confusing, and should be avoided in favour of a more understandable alternative terminology.

3. The response you gave to Reviewer 3’s point 1 could usefully be incorporated (in a briefer and more general way) into the paper as further explanation for your experimental design, perhaps in the legend to new Figure 1?

4. Please ensure that the highest quality figures are included in the final submission, since the versions I received in the original and revised submissions looked fuzzy/jagged/pixilated.